# Stem Cell Therapies in Movement Disorders: Lessons from Clinical Trials

**DOI:** 10.3390/biomedicines11020505

**Published:** 2023-02-09

**Authors:** Luca Marsili, Jennifer Sharma, Tiago Fleming Outeiro, Carlo Colosimo

**Affiliations:** 1Gardner Family Center for Parkinson’s Disease and Movement Disorders, Department of Neurology, University of Cincinnati, Cincinnati, OH 45219, USA; 2Department of Neurology, Queens University, Kingston K7L 3N6, ON, Canada; 3Department of Experimental Neurodegeneration, Center for Biostructural Imaging of Neurodegeneration, University Medical Center Göttingen, 37075 Göttingen, Germany; 4Max Planck Institute for Multidisciplinary Sciences, 37075 Göttingen, Germany; 5Translational and Clinical Research Institute, Faculty of Medical Sciences, Newcastle University, Framlington Place, Newcastle Upon Tyne NE2 4HH, UK; 6Deutsches Zentrum für Neurodegenerative Erkrankungen (DZNE), 37075 Göttingen, Germany; 7Department of Neurology, Santa Maria University Hospital, 05100 Terni, Italy

**Keywords:** movement disorders, stem cell therapies, neurodegeneration, disease-modifying therapies, precision medicine

## Abstract

Stem cell-based therapies (SCT) to treat neurodegenerative disorders have promise but clinical trials have only recently begun, and results are not expected for several years. While most SCTs largely lead to a symptomatic therapeutic effect by replacing lost cell types, there may also be disease-modifying therapeutic effects. In fact, SCT may complement a multi-drug, subtype-specific therapeutic approach, consistent with the idea of precision medicine, which matches molecular therapies to biological subtypes of disease. In this narrative review, we examine published and ongoing trials in SCT in Parkinson’s Disease, atypical parkinsonian disorders, Huntington’s disease, amyotrophic lateral sclerosis, and spinocerebellar ataxia in humans. We discuss the benefits and pitfalls of using this treatment approach within the spectrum of disease-modification efforts in neurodegenerative diseases. SCT may hold greater promise in the treatment of neurodegenerative disorders, but much research is required to determine the feasibility, safety, and efficacy of these complementary aims of therapeutic efforts.

## 1. Introduction

Neurodegenerative disorders result from a complex interplay between genes and environment. Due to their complexity and our limited understanding of the underlying pathological mechanisms, modelling these diseases in the laboratory has proven to be extremely challenging. Many cell and animal models have provided important mechanistic insight into neurodegenerative diseases but, thus far, there is a disconnect between therapeutic successes in animal models and those in clinical trials in humans.

Human stem cells (pluripotent and multipotent) are increasingly being studied as models or therapies for human disease. In most cases, stem-cell based therapies (SCTs) for neurodegenerative disorders rely on replacing lost cell types (e.g., replacing degenerated cells with new ones), thus exerting a symptomatic therapeutic effect. However, while cell replacement may provide rescue and neurorestorative effects, it is likely that disease-modification should rely on precision medicine approaches, matching molecular therapies to biological subtypes of disease. Within the precision medicine approach, which involves combination of multi-drug treatments, rather than a monotherapy, SCT may play a significant contribution in the treatment of neurodegenerative disorders [1]. 

Over the years, there has been significant hope that SCT would lead to curative therapies in these diseases, but unfortunately, that has not been observed thus far. We propose that, unless it addresses the inciting etiology, which is expected to vary among affected individuals, it will never be completely curative. However, with the recent development of newer and more effective cell lineages, differentiation processes, and grafting techniques, the once-imagined regenerative utopia may still be possible. Many review articles have been written on the possible utilization of stem cell therapies in various animal models and/or in patients with dementia, but very few have specifically discussed this topic in patients with movement disorders [2,3,4,5,6]. This narrative review will cover the basics of SCT in neurodegenerative movement disorders, such as Parkinson’s Disease (PD), atypical parkinsonian disorders (APD), Huntington’s disease (HD), amyotrophic lateral sclerosis (ALS), and spinocerebellar ataxia (SCA). We will also discuss the benefits and pitfalls of using this approach. 

Stem cells can be classified on their intrinsic ability to differentiate into the end organism [7]. There are five main categories of stem cells: totipotent, pluripotent, multipotent, oligopotent, and unipotent (**Glossary**). Stem cells can also be categorized as embryonic stem cells (ESCs)—cells derived during early development—and adult stem cells— rare, undifferentiated cells present in many adult tissues [8]. Special attention has been given to pluripotent ESCs, which can differentiate into any embryonic cell; initial trials required harvesting it at the blastocyte stage, but, in 2007, induced adult pluripotent stem cells (iPSCs) were artificially reprogrammed back from human fibroblasts or blood cells [9]. The ability to develop iPSCs, a non-embryonic source of multipurposed cells, was a breakthrough that avoided many ethical pitfalls as they can be derived from the patient’s own cells (e.g., autologous stem cells), and they avoid the risk of immunological rejections that are associated with non-autologous or heterologous stem cells [10]. Mesenchymal cells derived from the mesoderm and neuroectoderm were initially obtained from bone marrow; common sources now include adipose tissue, placenta, and umbilical cord, and they have the ability to differentiate into cell types from all three embryonic layers [11]. Interestingly, they can grow towards inflammation through the expression of chemokine receptors, making it an attractive candidate for cell loss, secondary to inflammatory conditions [12]. Totipotent cells are infrequently used in research as they are difficult to isolate and, once again, ethical questions arise. The above provides a simplified description of these categories to understand the following concepts.

## 2. Methods 

In the present narrative review, we searched PubMed until 10 November 2022, to identify key articles to screen for main results and for relevant bibliography. We mainly focused on the more informative studies in humans, with valid and clear methodology, and with the more recent dates of publication, if multiple similar studies were available. Single case reports were not considered, unless otherwise specified. Particular attention was given to multiple publications from the same trial with different follow-up periods. 

In Parkinson’s disease (PD), we used the terms “Parkinsonism” OR “Parkinson’s Disease”, as well as “Stem Cell” OR “Allogenic” OR “autologous” OR “Transplantation”. We also applied the same search terms to search in ‘www.clincaltrial.gov’, and we selected “ongoing trials”. The search terms were purposely non-specific to allow for a greater number of results so that they could be filtered individually. This resulted in 60 trials. A large number were removed for non-stem cell related mechanisms, fecal transplants, other neurodegenerative conditions, and incomplete and self-terminated studies. This resulted in 15 studies (Table 1). In multiple system atrophy (MSA), progressive supranuclear palsy (PSP), cortico-basal syndrome (CBS), and dementia with Lewy bodies (DLB) we used the following search terms: “Multiple Systems Atrophy”, “MSA”, “Progressive Supranuclear Palsy”, “PSP”, “Corticobasal Degeneration”, “Corticobasal syndrome”, “CBD”, “CBS”, and “stem cell OR autologous OR transplant”. In HD we used the following search terms: “Huntington’s Disease” OR “Huntington’s Chorea” and “stem cell OR autologous OR transplant”. In ALS we used the following search terms: “ALS”, “Amyotrophic Lateral Sclerosis”, “motor neuron disease”, and “stem cell OR autologous OR transplant”. In SCA we used the following search terms: “spinocerebellar ataxia”, “SCA”, and “stem cell OR autologous OR transplant”.

## 3. Parkinson’s Disease 

PD is a neurodegenerative syndrome which results in a loss of dopaminergic neurons, leading to nigrostriatal degeneration [17]. The pathogenesis of neuronal degeneration in PD likely involves the polymerization of alpha-synuclein, with a subsequent loss of normal, soluble synuclein and degeneration. As such, PD belongs to the category of “synucleinopathies” [18]. While serotonin and acetylcholine are involved to some extent, the mainstay of therapy has always been and continues to be dopamine replacement [19]. The loss of dopaminergic neurons is mainly located in the substantia nigra (SN)-*pars compacta* and its projections to the striatum. SCT cannot address the disease-causative mechanisms but can replace dopaminergic-producing cells. 

Stem cell transplants in PD started in the mid-1990s, with variable results. Olanow et al. showed that fetal stem cell transplants could improve motor symptoms (as measured by the Unified Parkinson’s Disease Rating Scale (UPDRS)-part III) up to 9 months after the transplant, and that this effect was not maintained at 12 or 24 months (e.g., primary endpoint not met) [14]. This transient improvement in UPDRS scores coincided with the duration of immune-suppressant post-transplant, suggesting a loss of the transplanted tissue. This was more evident in patients with somewhat less severe PD (UPDRS-III score < 49) when compared to those with more advanced PD (UPDRS-III score > 49). Overall, PD patients in early SCT trials had a range of responses; from no response, disabling dyskinesias (mainly due to heterogeneous fetal grafts containing both dopaminergic and serotonergic cells) to discontinuation of oral levodopa medication, it was hard to predict how patients would respond [20]. Failure of these trials was attributed to factors like including non-motor predominant patients, insufficient amount of transplanted tissue, older age, and more diffuse loss of dopaminergic neurons. Those who had dopamine neuronal loss, restricted to putamen, benefited more from this treatment [21]. A few encouraging single case reports have suggested that if the graft survives after immune-suppression discontinuation, and patients are properly selected, the effects of transplanted dopaminergic-producing cells could be tangible up to 20 years after the transplant [22,23]. However, these are observations based on single cases and it is hard to generalize their effect. Additionally, after decades of failures of SCT in PD [13,15,16,21] (Table 1), experts tried to review and devise new strategies for clinical trials. TRANSNEURO is a current, ongoing multicenter trial that involves implantation of allogenic human iPSCs-derived into the putamen and addresses some of these past limitations [16]. Preliminary data at 36 months on 11 subjects suggests the absence of disabling dyskinesias, continued deterioration of motor signs per Movement Disorders Society Unified Parkinson’s Disease Rating Scale (MDS-UPDRS) part III, lack of evidence for association between disease duration and clinical outcomes, and no major cognitive problems [16]. Separately, Aspen Neuroscience (ANPD001) has started recruitment for their autologous, iPSCs-derived SCT for idiopathic PD. It will avoid the need for immunosuppressants but does come at the cost of developing personalized individual cell lines for each individual. The feasibility of this approach has already been demonstrated in a single patient in whom PET imaging showed graft survival and a 6% reduction of levodopa requirement at 24 months [24]. 

Currently, there are fifteen ongoing clinical trials on parkinsonism and SCT (Table 1). Out of these, thirteen deal specifically with PD, suggesting that momentum continues within this field. Limitations have included choosing the right stem cell source, creating a cost-effective process to derive cell lineage in sufficient quantities, proper patient selection (currently restricted to motor-predominant and levodopa-responsive individuals in “earlier” stages), appropriate placement of the graft with verification of synapse connection to host networks, and finally, ensuring the longevity of grafts. Of interest, ongoing pre-clinical and phase I trials are mainly using iPSCs or ESC, whereas phase II trials also use mesenchymal stem cells (MSC).

## 4. Atypical Parkinsonian Disorders

APD are a broad number of conditions with PD-like phenotypes and include MSA, PSP, CBS, and DLB [25]. APD are primarily characterized by the combination of parkinsonism with additional motor and non-motor features that are beyond the “classical” spectrum of idiopathic PD, with a more aggressive disease progression. From a pathophysiological standpoint, APD consist of the “synucleinopaties”, such as MSA and DLB (with the common hallmark of soluble α-synuclein loss with corresponding insoluble α-synuclein accumulation), and “tauopathies” (characterized by soluble tau loss with corresponding insoluble tau accumulation), which include PSP and CBS [18,25]. The main studies exploring the effects of stem cells in APD are presented in Table 2.

*MSA*. Studies on stem cells in MSA patients have suggested a putative transient disease-modifying role of stem cells [30]. However, these studies have been conducted in a small number of patients (mainly MSA-cerebellar subtype and not on MSA-parkinsonian subtype) and in single centers, and without a double-blinded approach, implanting intravenous or intraarterial MSC [26,30]. These studies have not been followed by others on wider number of patients and other MSA subtypes; additionally, some of these studies have been published more than a decade ago without further confirmation studies [26], thus suggesting limited applications of stem cells for MSA patients. Furthermore, as multiple systems and cell types are affected in MSA, different cell types may be needed for a stem-cell based therapeutic approach in this condition [31].

*PSP*. It has been documented in PSP that bone marrow MSC can be safely used with a possible beneficial effect (or, at least, with stabilization of disease progression), after having excluded the placebo effect [27]. The rationale of MSC in PSP is not to replace diseased neurons, but rather to minimize the consequences of neural cell deterioration by using stem cells as treatment [28]. Single-case reports have documented encouraging results with intraarterial autologous adipose tissue-derived MSC [32] and umbilical cord blood stem cell transplantation [33] in patients with PSP.

*CBS and DLB.* Studies specifically conducted in patients with CBS and DLB are lacking. We only found a case series describing the outcomes of the intravenous administration of granulocyte colony stimulating factor (GCSF) (which stimulates the differentiation of hematopoietic stem cells) in patients with MSA, PSP, and CBS (*n* = 2). Patients with CBS showed improvement (*n* = 1) or stability (*n* = 1) in motor scales over the study period (3 months), but no follow-up was available [29]. 

## 5. Huntington’s Disease

HD is a neurodegenerative autosomal dominant condition associated with a CAG repeat expansion in the *HTT* gene, with the number of repeats shown to be inversely correlated with the disease severity and age at onset [34]. However, the recent failure of trials of the antisense oligonucleotide tominersen, aimed at reducing the “toxic” level of huntingtin (HTT) protein, possibly suggests that the “causal” role of HTT in this monogenic disease is hypothetical, and thus, that further studies are required to determine the neurodegenerative mechanisms associated with HD [35].

The clinical spectrum of HD includes motor (chorea, dystonia, ataxia) and cognitive-behavioral symptoms, due to a dysfunction in the striatum [36]. Only symptomatic treatments are available for HD. Some studies have tested the role of intrastriatal transplantation of fetal striatal neuroblast cells as a possible symptomatic, and in some cases, even putative rescue/neurorestorative disease-modifying treatment, showing some promising results at different follow-up periods [37,38,39,40,41,42,43] (Table 3). Nevertheless, as in APD, these open-label single-center studies were not followed by other trials on a broader number of participants, thus bringing into question the real effectiveness of their therapeutic approach. However, most studies have documented the overall safety of the transplant procedures in HD patients. 

## 6. Amyotrophic Lateral Sclerosis

Amyotrophic lateral sclerosis (ALS) is a progressive neurodegenerative disorder characterized by the dysfunction of both upper and lower motor neurons, leading to muscle weakness and paralysis, and eventual death [44]. Studies in ALS models have suggested that the primary pathophysiological target should be the environment of the motor neuron rather than the motor neuron itself [45]. Studies with different stem cell types in ALS patients have documented variable results over years, but overall, stem cell transplantation may delay ALS progression, improving quality of life (Table 4) [46,47,48,49,50,51,52]. Although multiple molecules (e.g., Vascular endothelial growth factors—VEGF, angiopoietin-related growth factor—ANG, and transforming growth factor beta—TGF-β) have been investigated, we are still missing effective biological markers to predict the efficacy of MSC transplants in patients with ALS [53]. Importantly, female sex and a positive therapeutic response to the first stem cell infusion are predictors of the efficacy of treatment with MSC in ALS patients [54]. Interestingly, the rate of disease progression seems to be an important predictor of therapeutic response to MSC in ALS patients, meaning rapid progressors have a better therapeutic response [55]. A recent study has published preliminary results on 18 ALS patients implanted, in the spinal cord, with human neural progenitor cells transduced with GDNF, because animal studies showed their ability to differentiate to astrocytes, thus protecting motor neurons [56]. These preliminary data showed encouraging results in terms of safety and tolerability at 12 months in ALS patients; however, and more interestingly, data from autoptic studies in a subgroup of patients suggested graft survival and satisfactory GDNF production. In sum, the future applications of gene therapy, combined with stem cell infusion, will consist of bilateral cortical and spinal cord infusions of STC, transduced with GDNF, to have a trophic effect for first and second order motor neurons [57]. 

## 7. Spinocerebellar Ataxia

SCA consists of a heterogeneous group of autosomal-dominant neurodegenerative conditions associated with ataxia, often caused by exonic CAG trinucleotide repeat expansions, which cause long polyglutamine chains [34,58]. More than forty types of SCA have been described up to date, many of them with associated movement disorders [58]. Among possible disease-modifying therapies, treatments with MSC have been proposed for SCA with the idea of stimulating plasticity and cell differentiation in the cerebellum [59]. However, only a few studies have explicitly tested MSC in patients with SCA, with limited evidence [60,61,62] (Table 5). Based on the results of a recent systematic review and meta-analysis of stem cell therapy in SCA, we can say that there is low evidence for recommending stem cells for these heterogeneous conditions, and that trials with large sample sizes and a lower risk of bias are still missing [63]. 

## 8. Current Challenges and Future Directions 

In the present review, we have performed a narrative review of the known studies in human stem cell transplantation for patients with neurodegenerative movement disorders. Many of the examined trials have investigated the role of SCT for symptomatic treatment, while other studies have tried to investigate the possible disease-modifying role of SCT through the production of growing/trophic factors.

The limited results of these trials can be attributed to multiple issues. First, the complexity of the various diseases makes it difficult to generate animal models that accurately reproduce the full diversity of disease features. Studies on animal models using iPSC have been useful to address early-onset neurodevelopmental diseases (e.g., Rett or Dravet syndromes, spinal muscular atrophy, or Friedreich’s ataxia), but not late-onset neurodegenerative disorders, which are the focus of this review. This is due to intrinsic problems of iPSC, namely the lack of maturation of the iPSC-derived cell lines [64]. Additionally, the existing iPSC models are only able to reproduce cells that show the behavior and features of the fetal cells. Aging should be considered as an independent factor to allow the in vitro modeling of the late-onset neurodegenerative diseases [65]. An accurate reproduction of late-onset neurodegeneration is as difficult as it is challenging to physiologically induce ‘aging’ into these animal models. [65]. Lastly, animal models are models of a disease construct, as we have defined it, not of people affected (e.g., PD patients, HD patients, etc.). In the last decades, the scientific field has significantly progressed, thus offering a great knowledge about disease mechanisms, but very little is known about the molecular biology of people affected by these diseases. Hence, we should be aware that animal models may not always be suitable to keep testing our hypotheses. 

Another challenge in SCT is delivering the cells to their target sites within the central nervous system (CNS) in a safe and consistent way. Peripheral delivery of SC through intravenous infusions can lead to sequestration in tissues and organs, like the spleen, liver, and lung, thus preventing their passage through the blood brain barrier (BBB) [66,67]. Intraarterial injections are more effective for neurological conditions [68], but this increased efficacy comes at the cost of potential safety issues, such as microvascular complications and higher mortality [69]. Other trials are currently proposing CT/MRI-guided neurosurgical injections of STC that have the advantage of reaching the target structures precisely, and less adverse events (mainly hemorrhages). On this basis, a *n* = 1 patient-funded, compassionate use personalized trial with autologous iPSCs has been recently conceived, and preliminary clinical results at 24 months are highly encouraging. As they require substantial resources, SCT trials have commonly occurred in single centers where expertise and resources can be concentrated. Recent ongoing trials, such as the TRANSNEURO, have attempted to account for this and they have illuminated the challenges faced by multiple trials, primarily with recruitment and timing stem cell obtainment and implantation [16].

To partially overcome these limitations, an innovative phase I/II clinical trial, using human ESC-derived midbrain dopamine neurons (MSK-DA01) for patients with advanced PD and older age (>74) [70], has recently been approved by the FDA (Dec 2020; NCT04802733). The results are not yet available. As with many trials in neurodegeneration, patient selection and the measurement of disease modification is of utmost importance. Many studies have only examined SCT in earlier stages of PD, in a certain phenotypic subtype of the disease in MSA (e.g., MSA-C versus MSA-P), or in APD, in the very advanced stages, as a “rescue” strategy in patients who are already in poor health. We believe that the correct selection of patients should not be based on clinical/phenotypical criteria, but rather on biological and molecular ones. The highly encouraging results of clinical trials in ALS have proven that female sex, rapid disease progression, and positive response to the first SCT infusion are good predictors of effectiveness [54,58]. Additionally, patients with motor-predominant PD and dopaminergic loss, confined to the putamen, tend to respond much better to SCT than those without these features [20]. True disease-modification effect is hard to measure in neurodegenerative disease due to the lack of precise biomarkers [71]. Moreover, it is difficult to distinguish between an early symptomatic therapeutic effect and a true disease-modifying effect, at least in early (untreated) disease stages [72]. The tested endpoints may be highly variable and include the time-to-milestones of disease progression, the change in the MDS-UPDRS (or other clinical scores), and modification in a given imaging biomarker of metabolic function (e.g., dopaminergic in parkinsonian syndromes) [72,73]. Future disease-modifying trials may abandon the subjective clinical scales and questionnaires in favor of the objective evaluations offered by technology-based objective measures, imaging, and/or new molecular markers [71,73,74,75].

A final consideration for SCT is related to their ethical and social aspects. In 2016, the International Society for Stem Cell Research (ISSCR) created a guideline to regulate the rigorous scientific inquiry and careful ethical deliberations regarding SCT and regenerative medicine [76]. These guidelines contain the ethical principles for guiding both basic and clinical SCT research by regulating and codifying the integrity of research, patients’ welfare, respect for research participants, transparency, and finally, social justice. The discovery of iPSCs helped avoid some of the previously raised issues, but these guidelines will continue to play a large role as induced totipotent cells became a reality [77]. Lastly, researchers are increasingly debating the ethical ramifications of performing sham procedures. This includes procedures, such as sham neurosurgical injections, in double blinded trials. There is a possibility that researchers may abandon this process, opting for open label, single arm trials. 

## 9. Final Remarks

Overall, interest in SCT has waxed and waned over more than thirty years, but there is a recent, significant renewed interest. We have since realized that the etiopathogenesis of neurodegenerative diseases is far more complex than previously thought and that each entity might not represent one unifying diagnosis, but rather, the gross pathology represents the end-product of various, unique etiologies [78]. Nevertheless, gold-standard diagnosis continues to rely on pathology [79,80]. Clinical and prognostic heterogeneity is the rule rather than the exception and affects patient selection and the ability of each body to accept the grafts. There is significant work underway in hopes of uncovering biomarkers that may better help identifying these various etiologies, perhaps leading to curative treatments in very select individuals. As displayed in the Figure 1, success in SCT will involve choosing the appropriate stem cell line from which to derive iPSCs, creating a streamlined, cost-effective, and scalable method in which to produce stem cells, ensuring safe delivery of stem cells past the BBB, in sufficient quantities, to the ideal location with synapses connection and growth, clinically relevant measurement of graft success, and lastly, appropriate patient selection and representation in trials. Despite the still limited knowledge on the mechanisms of neurodegeneration, we believe that future SCT may help in tackling some specific subtypes of neurodegenerative diseases. To this aim, we should move away from the old clinicopathology-based nosology of neurodegenerative disorders and focus on their underlining biological mechanisms [80,81]. It remains to be seen what future studies will uncover; if, indeed, we are able to accomplish the above, it may prove to be an ideal symptomatic therapy for certain individuals. As an example, around 10% of pathology-proven PD cases are unresponsive to dopaminergic treatment, and an additional 12% have a modest response [82]; these patients, to our knowledge, could be suitable candidates to symptomatic SCT based on dopaminergic neuronal transplantation. Other than that, SCT may also be useful for disease-modifying treatment options, which rely on precision medicine approaches. In fact, SCT may hold greater promise in the treatment of neurodegenerative disorders as part of a multi-drug, subtype-specific therapeutic approach, but much research is required to determine the feasibility, safety, and efficacy of these complementary aims of therapeutic efforts (Figure 1).

Upper panel: Stem cell therapies (SCT) may have both symptomatic and disease-modifying effects. The diagram also illustrates a major difference between symptomatic treatments, which are largely universal, versus disease-modifying treatments, which encompass both rescue and restorative treatments and precision medicine, which must be individualized, requiring biological subtyping. Lower panel: Stem cell therapies in the future should be based on biological markers of disease progression and administered through autologous transplantation, possibly directly injected into the brain target sites. This approach will hopefully allow cell regeneration and ultimately, a possible restorative disease-modifying effect improving patients’ symptoms. Created with BioRender.com.

## 10. Glossary

**Totipotent cells** can differentiate into any cell in the body—both embryonic and extra-embryonic tissue, such as the placenta. In mammals, they consist of the zygote (fertilized egg) and the first blastomeres cells (up to stages 4–8 cells).**Pluripotent cells** include embryonic stem cells (ESCs) and induced pluripotent stem cells (iPSCs). Pluripotent cells can differentiate in any embryonic cell but cannot differentiate any extra-embryonic tissue. It can form all three germ layers and is formed from the inner cell mass of a blastocyst; these are specifically named embryonic stem cells (ESC).**Multipotent cells** can differentiate into a family of cells that, in many cases, belong to the same tissue (e.g., these cells can develop only into specialized cell types). Examples include Neural Stem Cells (NSCs), which can become any cell part of the central nervous system.**Oligopotent** cells can differentiate into a limited number of different cells.**Unipotent cells** can generate a single type of cell.

## Figures and Tables

**Figure 1 biomedicines-11-00505-f001:**
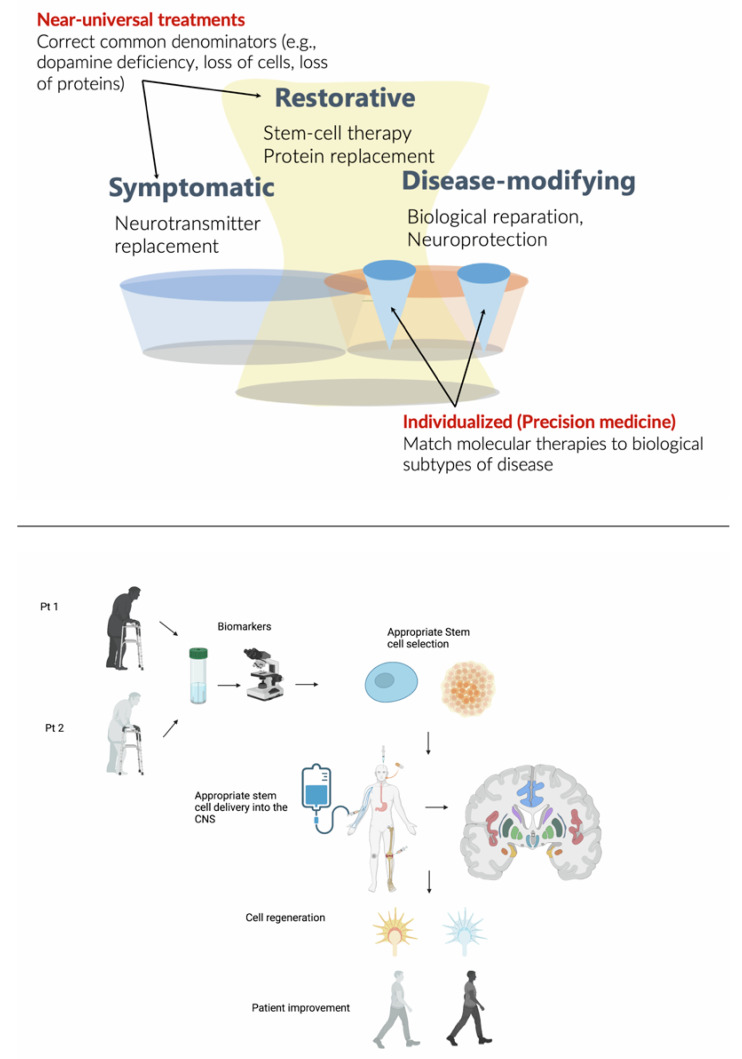
Schematic representation of the main theme of the review.

**Table 1 biomedicines-11-00505-t001:** Main published and current studies exploring stem cell treatments in patients with idiopathic Parkinson’s disease.

Condition	Study Name/ID	Phase	N of Patients	Intervention	Design	Status	Follow	Outcome
							Up	
PD	Freed et al., 2001 [13]	I	40	Putaminal allogenic MSC	Double-blind, placebo (real vs sham surgery), RCT	Published	12 m	Subjective clinical improvement in younger age, not in elderly. Dystonia and dyskinesias returned in 15% of transplanted patients at 12 m
PD	Olanow et al., 2003 [14]	I	34	Fetal nigral transplantation	Multidose,	Published	24 m	Primary endpoint not met (UPDRS-III change at 24 m)
					placebo-controlled, double-blind			
* PD	Ma et al., 2010 [15]	I	33	Putaminal allogenic MSC	Double-blind, placebo (real vs sham surgery), RCT	Published	48 m	The dependence of clinical outcomes on subject age and sex at 12 m may not persist at 48 m
PD	Barker et al., 2019 [16]	I	11	Putaminal allogenic MSC	Randomized, open label	Published	36 m	UPDRS decline (preliminary data), no cognitive disability
PD, MSA, MSA-P	NCT04876326	NA	15	Autologous adipose MSC, Allogenic umbilical cord MSC, both	Randomized, parallel assignments	Recruiting	-	Clinical, eyesight changes, imaging
PD	NCT03128450	II/III	12	Human SC	Single arm	Unknown	-	UPDRS, motor, non-motor functions, QoL
PD	NCT03550183	I	20	MSC	Single arm	Enrolling	-	UPDRS, MMSE, HAMD, HAMA
PD	NCT03684122	I/II	10	Umbilical cord MSC	Randomized, open label	Not recruiting	-	Safety, tractography, blood, CSF biomarkers
PD	NCT01446614	I/II	20	BM-MSC	Single arm, open label	Unknown	-	Safety
PDD, AD, APD	NCT03724136	NA	100	IV/IN BM-MSC	Non-randomized, parallel assignment open label	Recruiting	-	MMSE, ADL
PD	NCT04506073	II	45	MSC	Randomized, parallel assignment	Not recruiting	-	UPDRS, safety, TUG, H&Y, ADL, PDQ-39, QoL, MoCA
PD	NCT04146519	II/III	50	Autologous MSC	Randomized, parallel assignment	Recruiting	-	Motor, non-motor symptoms, sleep quality, depression
PD	NCT04928287	II	24	MSC	Randomized placebo-controlled	Not recruiting	-	UPDRS, Safety, Lab values
PD	NCT04414813	I	3	Amniotic epithelial SC	Single arm, open label	Not recruiting	-	Safety, UPDRS, H&Y scale, PDQ-39
PD	NCT04414813	I	10	iPSC	Single arm	Unknown		Safety
PD	NCT02452723	I	12	SC	open label, single arm	Unknown	-	Safety, UPDRS
PD	NCT03119636	I/II	50	ESC-derived neural precursor cells	Non-randomized, open label	Unknown	-	Safety, UPDRS, DATscan, H&Y
PD	NCT04802733	I	12	ESC	Single arm, open label	Not recruiting	-	Safety, motor function

AD, Alzheimer’s disease; ADL, activities of daily life; APD, atypical parkinsonian disorders; BM-MSC, bone marrow derived mesenchymal stem cells; CSF, cerebrospinal fluid; DTBZ, [18F] 9-fluoropropyl- (+)-dihydrotetrabenazine; ESC, embryonic stem cells; FDOPA, L-3,4- dihydroxy-6-(18)F- fluorophenylalanine; H&Y, Hoehn and Yahr scale; HAMA, Hamilton anxiety rating scale; HAMD, Hamilton depression rating scale; IN intranasal; iPSC, induced pluripotent stem cells; IV, intravenous; m, months; MMSE, mini-mental state evaluation; MoCA, Montreal cognitive assessment; MSC, mesenchymal stem cells; N, number; NA, not applicable; NA, not assessed; PD, Parkinson’s disease; PDD, Parkinson’s disease dementia; PDQ-39, Parkinson’s Disease Questionnaire 39; QoL, quality of life; SC, stem cells; TUG, time-up-and-go; UPDRS, Unified Parkinson’s Disease Rating Scale. * This is the follow-up to the study by Freed et al., 2001 [13].

**Table 2 biomedicines-11-00505-t002:** Main studies exploring stem cell treatments in patients with atypical parkinsonian syndromes.

Condition	Study Name	Phase	Number of Patients	Intervention	Design	Adverse Events	Follow-Up	Outcome
MSA-C	Lee et al., 2012 [26]	I	33	IA or IV MSC	2 arms vs. placebo	Small ischemic lesions in IA	12 m	Slower decrease in UMSARS
PSP-RS	Giordano et al., 2014 [27]	I	25	IA BM-MSC	Placebo-controlled crossover	Small ischemic lesions	18 m	Slower decrease in UPDRS; increased FDG-PET uptake
PSP	Canesi et al., 2016 [28]	I	5	IA BM-MSC	1 arm	NA	12 m	Stable rating scales
MSA (*n* = 4), PSP (*n* = 5), CBS (*n* = 2)	Pezzoli et al., 2008 [29]	I	11	IV GCSF	1 arm	None	3 m	UPDRS not worsened significantly

BM-MSC, bone marrow-derived mesenchymal stem cells; CBS, corticobasal syndrome; GCSF, granulocytes colony-stimulating factors; IA, intra-arterial; IV, intravenous; m, months; MSA-C, MSA-cerebellar type; MSA, multiple system atrophy; MSC, mesenchymal stem cells; n, number; NA, not assessed; PSP-RS, PSP-Richardson’s type; PSP, progressive supranuclear palsy; RS, Richardson syndrome; UMSARS, Unified MSA Rating Scale. Single case reports are not included in the present table.

**Table 3 biomedicines-11-00505-t003:** Main studies exploring stem cell treatments in patients with Huntington’s disease.

Study Name	Phase	Number of Patients	Intervention	Design	Follow-Up	Adverse Events	Outcome
Bachoud-Lévi et al., 1999 [43]	I	5	Fetal striatal neural allografts	1 arm, open label	12 m	None	Stable motor and cognitive symptoms
Bachoud-Lévi et al., 2000* [37]	I	5	Intrastriatal neuroblasts	1 arm, open label	12 m	NA	Slower progression of motor and cognitive symptoms
Hauser et al., 2002 [39]	I	7	Fetal striatal tissue transplantation	1 arm, open label	12 m	Subdural hemorrhages (*n* = 3)	Slower progression in UHDRS
Bachoud-Lévi et al., 2006 [38] *	I	5	Intrastriatal neuroblasts	1 arm, open label	72 m	NA	Clinical improvement plateaued at 2 y, then faded off at 4–6 y
Reuter et al., 2008 [40]	I	2	Fetal striatal allografts	1 arm, open label	66 m	NA	Improvement in UHDRS, cognition, and mood
Barker et al., 2012 [42]	I	5	Fetal striatal tissue transplantation	2 arms: control group	54.6 m (AV.)	None	No difference between groups
Paganini et al., 2014 [41]	I	10	Fetal striatal tissue transplantation	2 arms: control group	132 m (Av.)	Not relevant	Lower motor, cognitive progression; better brain metabolism in transplanted pts.

Av., Average; HD, Huntington’s disease; NA, not assessed; pts., patients; UHDRS, unified HD rating scale; y, years. * Same trial but with different follow-up periods.

**Table 4 biomedicines-11-00505-t004:** Main studies exploring stem cell treatments in patients with amyotrophic lateral sclerosis.

Study Name	Phase	N of Patients	Intervention	Design	Follow-Up	Adverse Events	Outcome
Deda et al., 2009 [47]	I	13	BM-derived hematopoietic SC	1 arm, open label, control group	12 m	Infections	Global motor improvement (ENMG)
Martinez et al., 2010 [46]	I	10	CD133+ SC	1 arm, open label, control group	12 m	None	Transient increase in ALSFRS-R score (higher better) at 6 m, and improved survival
Glass et al., 2012 [48]	I	12	Intraspinal fetal-derived neural SC	1 arm, open label	18 m	None	No evidence of acceleration of disease progression at ALSFRS-R, FVC, HHD scales
Riley et al., 2012 [50]	I	12	Intraspinal neural fetal SC	1 arm, open label	18 m	Surgery-related	Intraspinal lumbar microinjection procedure is safe (Safety trial)
Mazzini et al., 2019 [51]	I	18	Intraspinal neural SC	1 arm, open label	60 m	None	Transient increase in ALSFRS-R scale at 1 and 4 m
Barczewsk a et al., 2020 [54]	I	67	WJ-MSC	1 arm, open label, and control group	6 m	None	Increased survival
Siwek et al., 2020 [55]	I	8	BM-MSC	1 arm, open label	6 m	None	Slowing of disease progression (ALSFRS-R score) only in patients with rapid disease course
Petrou et al.l., 2021 [52]	II	20	MSC	1 arm, open label	6 m	None	>25% improvement in ALSFRS-R
Baloh et al., 2022 [56]	I/IIa	18	Unilateral spinal injection of human progenitor neural SC transduced with GDNF	1 arm open label	12 m	None	Procedure is safe (Safety trial); Graft survival and GDNF production were confirmed in autopsied samples

ALSFRS-R, ALS functional rating scale-revised; BM, bone marrow; ENMG, electroneuromyography; FVC, forced vital capacity; HHD, hand-held dynamometry; MSC, mesenchymal stem cells; SC, stem cells; WJ, Wharton’s jelly.

**Table 5 biomedicines-11-00505-t005:** Main studies exploring stem cell treatments in patients with spinocerebellar ataxia.

Study Name	Phase	Number of Patients	Intervention	Design	Adverse Events	Follow-Up	Outcome
Dongmei et al., 2011 [60]	I	24 *	Intrathecal injection of UC-MSC	1 arm, open label	None	15 m	Better ICARS and ADL scores
Jin et al., 2013 [61]	I	16	IV and intrathecal UC-MSC	1 arm, open label	None	12 m	Better BBS and ICARS scores
Tsai et al., 2017 [62]	I/IIa	7 **	IV allogeneic adipose tissue derived MSC	1 arm, open label	None	12 m	Marginally better SARA and PET metabolism

ADL, activity of daily living; BBS, Berg balance test; ICARS, international cooperative ataxia rating scale; IV, intravenous; SARA, scale for the assessment and rating of ataxia; UC-MSC, umbilical cord mesenchymal stromal cells. * Included 10 patients with spinocerebellar ataxia and 14 patients with multiple system atrophy type C. ** Included 6 patients with spinocerebellar ataxia type 3 and 1 with multiple system atrophy type C.

## Data Availability

Not applicable.

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
