# Peer review of "Stem Cell Therapies in Movement Disorders: Lessons from Clinical Trials"

_biomedicines, 2023, doi:10.3390/biomedicines11020505_

Round 1

Reviewer 1 Report

The review is very interesting and the theme is an emerging field in the era of research.

However some major issues emerged after reading the manuscript.

MAJOR ISSUES

The search criteria are not stated. Were the authors performed the research of clinical trials? What are the keywords used? What databases were used?

A methods section is warranted.

Line 57: add a couple of references/examples.

Line 170, 180, 188: add more details. How were MSC implanted?

Figure is not informative. Probably the aim of the authors was to highlight the problem related with such kind of therapy. Is so, more details should be added in the figure. Otherwise, it could be removed.

MINOR ISSUES

Line 89: use subsequent instead of consequent.

Line 90 and 155: correct synucleinopaties with synucleinopathies

Line 102 and 103: please double check the symbol > and <.

Table 1: the acronyms in the footnote should be reported in alphabetical order.

Table 2: the acronyms in the footnote should be reported in alphabetical order.

Table 3: the acronym sin the footnote should be reported in alphabetical order.

Table 4: the acronyms in the footnote should be reported in alphabetical order.

Table 5: the acronyms in the footnote should be reported in alphabetical order.

Line 280: avoid the use of “a final comment of the authors”.

Line 282: what are the authors referring about, what we have learned about?

Author Response

Reviewer#1

 Comment: The review is very interesting and the theme is an emerging field in the era of research.

However some major issues emerged after reading the manuscript.

            Answer: We thank the Reviewer for the general positive comment.

 MAJOR ISSUES

Comment: The search criteria are not stated. Were the authors performed the research of clinical trials? What are the keywords used? What databases were used?

A methods section is warranted.

            Answer: We thank the Reviewer for this comment. Accordingly, we have now added a new methods section as required (Lines: 85-108). We have also better specified that our study was a narrative and not systematic review (Lines: 26, 60, 86). Given the narrative and not-systematic design of the review, we did not perform a systematic revision of the literature according to specific guidelines. However, we used specific search terms that were applied to search for the main contributive articles and studies on the topic. The reference section of each included study was then used to find further relevant articles. The only systematic search that we applied was related to the clinical trials on the use of SCT in Parkinson’ s disease. We have now added the search terms in the methods section (Lines: 92-99).

 Comment: Line 57: add a couple of references/examples.

            Answer: We agree with the Reviewer’s comment, and accordingly we have added some new supporting references (New references 2-6).

Comment: Line 170, 180, 188: add more details. How were MSC implanted?

            Answer: We have now better specified in the text as well in the related Table 2, the administration techniques of stem cells in the different reported studies (Lines: 195-196; 203 and 208; 212).

Comment: Figure is not informative. Probably the aim of the authors was to highlight the problem related with such kind of therapy. Is so, more details should be added in the figure. Otherwise, it could be removed.

            Answer: We thank the Reviewer for making this point clear to us. Accordingly, we have better organized the figure adding a second panel. The current figure now displays the main theme of the review and helps the readers in better understanding the differences between disease-modifying and neurorestorative approaches (upper panel), and to have a global idea on how stem cells should be created, and then administered to patients in the future (lower panel).

MINOR ISSUES

Comment: Line 89: use subsequent instead of consequent.

            Answer: We have now corrected the sentence (Line 113)

Comment: Line 90 and 155: correct synucleinopaties with synucleinopathies

            Answer: We have now corrected the sentence (Lines 114 and 177)

Comment: Line 102 and 103: please double check the symbol > and <.

            Answer: checked and corrected the related sentence (Lines 126-127)

Comment:

Table 1: the acronyms in the footnote should be reported in alphabetical order.

Table 2: the acronyms in the footnote should be reported in alphabetical order.

Table 3: the acronym sin the footnote should be reported in alphabetical order.

Table 4: the acronyms in the footnote should be reported in alphabetical order.

Table 5: the acronyms in the footnote should be reported in alphabetical order.

            Answer: We have now ordered the footnotes of the Tables in alphabetical order as suggested.

Comment:  Line 280: avoid the use of “a final comment of the authors”.

            Answer: It has been now removed (Line 302)

Comment: Line 282: what are the authors referring about, what we have learned about?

            We have now rephrased the sentence (Lines 304-306).

Reviewer 2 Report

Thank you for the opportunity to review the manuscript entitled: Stem cell therapies in movement disorders: Lessons from clinical trials

This is an interesting topic with a highly quality and relevance for movement disorders specialists and doctors treating these conditions.

The research topic is of scientific and social interest.

The topic is threated in correct form.

In general, the article is very interesting, clarifies the current moment of stem cells in research and medical field. Additionally, I consider that the topic is in line with the journal’s research objectives.

The conclusions and the discussion are well drawn and interesting.

I just wanted to suggest in lines 224 and 225 to clarify what the acronyms stand for, since it is the first time mentioned in the paper.

Additionally, please mention the figure in the body of the paper adding a small description explaining the purpose of it; inviting readers to consider the revision of the graph in detail. It seems to be a little secluded from the body of the text.

An interesting topic of research is explained

Congratulations to the authors

Author Response

Reviewer#2

Comment: This is an interesting topic with a highly quality and relevance for movement disorders specialists and doctors treating these conditions.

The research topic is of scientific and social interest.

The topic is threated in correct form.

In general, the article is very interesting, clarifies the current moment of stem cells in research and medical field. Additionally, I consider that the topic is in line with the journal’s research objectives.

The conclusions and the discussion are well drawn and interesting. 

Answer: We thank the Reviewer for the positive and encouraging comments.

Comment: I just wanted to suggest in lines 224 and 225 to clarify what the acronyms stand for, since it is the first time mentioned in the paper.

            Answer: We have now spelled out the meaning of the acronyms, as correctly suggested by the Reviewer (Lines 245-247).

Comment: Additionally, please mention the figure in the body of the paper adding a small description explaining the purpose of it; inviting readers to consider the revision of the graph in detail. It seems to be a little secluded from the body of the text.

            Answer: We thank the Reviewer for this comment. Accordingly, and also in line with Reviewer’s#1 comments, we have now better organized the figure adding a second panel. The current figure now displays the main theme of the review and helps the readers in better understanding the differences between disease-modifying and neurorestorative approaches (upper panel), and to have a global idea on how stem cells should be created, and then administered to patients in the future (lower panel). The text has been modified accordingly (Lines 368 and 387).

Comment: An interesting topic of research is explained

Congratulations to the authors

Answer: Again, we thank the Reviewer for the positive comments.